# Phosphorylated Metal–Organic Framework for Reducing Fire Hazards of Poly(Methyl Methacrylate)

**DOI:** 10.3390/polym14224871

**Published:** 2022-11-11

**Authors:** Lei Wang, Ximiao Hu, Zhelin Mao, Jianlei Wang, Xin Wang

**Affiliations:** 1College of Chemical Engineering and Material Science, Zhejiang University of Technology, Hangzhou 310014, China; 2Key Laboratory of Design and Assembly of Functional Nanostructures, Fujian Institute of Research on the Structure of Matter, Chinese Academy of Sciences, Fuzhou 350108, China

**Keywords:** flame retardant property, antistatic property, mechanical property, poly(methyl methacrylate)

## Abstract

The low fire safety performance (flame retardant and antistatic properties) of poly(methyl methacrylate) (PMMA) has severely limited practical applications. Here, a phosphorylated Zn-based metal–organic framework (ZIF-8-P) is employed as an effective flame retardant and antistatic agent to reduce the fire risk of PMMA. Encouragingly, the as-prepared PMMA/ZIF-8-P composite demonstrated not merely better mechanical properties (e.g., a rise of ca. 136.9% and 175.0% in the reduced modulus and hardness; a higher storage modulus), but also efficient fire safety properties (e.g., lower surface resistance; a decrease of ca. 73.1% in the peak heat release rate; a lower amount of total pyrolysis products), surpassing those of pure PMMA and a PMMA/ZIF-8 composite without phytic acid modification. Mechanism analysis is conducted to reveal the critical role of catalytic charring, char reinforcing, and the dilution of nonflammable gases from ZIF-8 additives during the combustion and pyrolysis process. Our study paves a promising way to achieve high performance PMMA composites.

## 1. Introduction

Poly(methyl methacrylate) (PMMA), also called as “acrylic glass” or “plexiglass”, has been widely used to replace true glass in various fields, finding use in aircraft and automotive windscreens, architectural applications, electronics, aquarium windows, and even medical implants due to the high transparency and impact and environmental resistance [1,2,3,4]. However, PMMA is very susceptible to fire, presenting potential safety hazards during practical applications. The detailed reason for the low fire safety property is described as follows. During the combustion process, a methyl methacrylate (MMA) monomer generated from PMMA, will further disintegrate into many small molecules made up of combustible materials (methanol, methane, acetone, and propylene) and emit a large amount of heat, which easily accelerates the combustion [5]. In addition, electrostatic accumulation is another disadvantage of PMMA, which could bring about adsorbing dust and even giving rise to an explosion in aerospace settings [6]. Consequently, it is highly urgent to develop new PMMA-based materials with excellent flame retardant and antistatic properties.

At present, adding agents is commonly used in the fire prevention of flammable polymers [7,8,9,10,11]. For example, inorganic-organic agents were reported as efficient additives to improve the performances of PMMA due to the excellent compatibility and dispersion state [12]. Metal organic frameworks (MOFs), as a typical inorganic-organic hybrid structure comprised of metal centers and organic ligands, have shown great potential in decreasing the flammability of polymers [13]. The advantages of MOF materials include the following two aspects: (1) superior compatibility with a polymer matrix [8]; (2) metal compounds from MOFs lead to a catalytic effect on promoting the char-forming and reducing smoke emission. ZIF-8, a Zn-containing zeolitic imidazole-based MOF with a rhombic dodecahedral structure, can be synthesized in large quantities by a simple method, which suits large-scale utilization well [14]. However, recent studies have revealed that the use of ZIF-8 alone as a flame-retardant additive does not work well. For instance, Shi et al. [15] reported that the PLA/1% ZIF-8 only reached a V-2 ratio in the UL-94 test.

In order to further improve the flame retardance of ZIF-8, the design of ZIF-8-based composite agents with other functional materials or individual components was proposed and showed some good results [16,17,18]. Wang et al. [17] reported that the addition of ZIF-8 with EG into a polyurethane elastomer (PUE) could bring about a high LOI value of 30.2% and a V-1 rating for UL-94 vertical burning test. Xu et al. [18] had shown that the LOI value of EP/ZIF-8/RGO (ZIF-8 loaded on the surface of reduced graphene oxide (RGO)) composite increased to 26.8% and V-1 rating was obtained for the vertical burning test (pure EP showed no UL-94 rating). Inspired by above analysis, the introduction of a proper ZIF-8-based compounded fire-retardant is expected to improve the combustion performance of PMMA. Moreover, some hydrophilic groups which can effectively prevent static electricity need to be introduced to improve the antistatic properties of PMMA [11]. The organic molecular chains in ZIF-8 are the gas and carbon sources [10]. ZIF-8 modified with biomass phytic acid (PA), a natural green material with abundant phosphorus (27.3% P), when applied to provide acid sources and hydrophilic groups, is expected to have higher efficiency regarding fire resistance and anti-static electricity [19]. Therefore, using ZIF-8-P is probably an ideal way to obtain green and high efficiency flame-retardant and anti-static properties in a PMMA composite.

In this work, we first design and synthesize hydrophilic biomass phytic acid-functionalized ZIF-8 (ZIF-8-PA), which is then used as an additive to prepare PMMA/ZIF-8-P composites via a facile in-situ polymerization method. Compared with pure PMMA and a PMMA/ZIF-8 composite, the as-made PMMA/ZIF-8-P composite displayed higher hardness and a reduced modulus value according to nanoindentation testing. Dynamic mechanical analysis (DMA) revealed that PMMA/ZIF-8-P exhibited the highest storage modulus below the glassy state region. Moreover, PMMA/ZIF-8-P also presented better flame retardancy (higher LOI value, lower PHRR data and less pyrolysis products) and antistatic properties (lower surface resistance value) than that of PMMA/ZIF-8. All the favorable features, including high mechanical resilience, flame retardance, and anti-static properties, offer such bulk materials great promise in special application fields.

## 2. Experiments

### 2.1. Materials

Methanol (CH_3_OH), 2-methylimidazole and zinc nitrate hexahydrate were purchased from Sinopharm Chemical Reagent Co., LTD (Shanghai, China). Phytic acid (PA, 50% (*w*/*w*) in H_2_O), methyl methacrylate (MMA) (99.5%, containing 30 ppm MEHQ stabilizer) and benzoyl peroxide (BPO) (C14H10O4) were supplied from Aladdin Biochemical Technology Co., LTD (Shanghai, China).

### 2.2. Preparation of ZIF-8-P

ZIF-8 was synthesized in the following steps as per the report [20]. First, 2.232 g of Zn(NO_3_)_2_·6H_2_O was dissolved in 60 mL of methanol and then 2.464 g of 2-methylimidazole was diluted in 60 mL of methanol. Then, the two solutions were mixed and stirred vigorously at room temperature for 6 h. The precipitated phase was centrifuged, washed with methanol several times, and then dried at 60 °C.

ZIF-8-P was synthesized in the following steps: First, 1.0 g of ZIF-8 was dispersed in 60 mL of deionized water. Then, 10.7 mL of PA was diluted in 20 mL of water. The two solutions were mixed for 12 h under magnetic stirring at room temperature. The powder was washed and collected by centrifugation several times and dried under vacuum at 80 °C for 48 h.

### 2.3. Preparation of PMMA and Its Composites

PMMA/ZIF-8-P was prepared by in-situ polymerization. MMA was mixed with acetone, where 0.1% initiator (BPO) was then added to the mixture. Afterwards, ZIF-8-P was immersed into the MMA/initiator mixture, which was stirred at 80 °C for about 1 h in an oil bath. When it was observed that the solution was about to form a condensate in the flask, the preliminary sample could be transferred to a plastic container. The reaction vessel was placed in an oven at 60 °C for polymerization and left in place for sufficient time to remove the incomplete volatilization of the liquid. The target product, a PMMA/ZIF-8-P nanocomposite, was then obtained. Here, the ratio of ZIF-8-P and PMMA was 1:99 by weight. The PMMA composite was prepared by the same method without ZIF-8-P. The PMMA/ZIF-8 composite was prepared with the method of in situ polymerization by directly adding ZIF-8 into the MMA/initiator mixture, and the weight ratio of ZIF-8 and PMMA was 1:99.

### 2.4. Characterization

Scanning electron microscopy (SEM) (JEOL-6700F) was used to investigate the morphologies of ZIF-8 and ZIF-8-P, as well as the dispersions of the prepared products and the images of the carbon slag after combustion. A thermogravimetric analyzer (TGA) (Perkin Elmer Instrument Shanghai Co., LTD., Shanghai, China) with a heating rate of 10 °C/min was adopted to characterize the thermal property of the samples. X-ray photoelectron spectroscopy (XPS) and X-ray diffraction (XRD) were employed to study the chemical constituents of the samples. A nanoindenter instrument (Tribo Indenter 750, Hysitron Inc., Eden Presley, MN, USA) was used to measure the micromechanical properties of samples. The dynamic mechanical properties of the samples were analyzed via dynamic thermomechanical analysis (DMA) (Perkin Elmer Diamond, Waltham, MA, USA) with a frequency of 5 Hz and a 5 °C/min heating rate. The antistatic property was tested by a surface resistance tester (ZC90F 150 V). The limiting oxygen index (LOI, HC-2 oxygen index instrument) of the samples can be used to analyze and evaluate the combustion performance of PMMA and its polymers according to the ASTM D 2863-77 standard. A vertical burning (UL-94) test was conducted with a CZF-II horizontal and vertical burning tester (Jiang Ning Analysis Instrument Co., Nanjing, China), according to the ASTM D 3801-1996 standard. Microscale combustion calorimetry (MCC) (Govmark) was performed to analyze the combustion properties of the samples according to ASTM D 7309-7. Thermogravimetric analysis including Fourier-transformed infrared spectrometry (TGA-FTIR) of the samples was employed to use the TGA Q5000 IR thermo-gravimetric analyzer that was interfaced to the Nicolet 6700 FT-IR spectrophotometer. Laser Raman spectrometry (LRS) measurement was carried out by a SPEX-1403 laser Raman spectrometer (SPEX Co., Metuchen, NJ, USA) with excitation provided in back-scattering geometry by a 514.5 nm argon laser line at room temperature.

## 3. Results and Discussion

Figure 1a,b display SEM images of the sample, in which the morphology of ZIF-8 presents as an regular octahedron. After further functionalization with PA, its morphology had some changes and did not show a regular morphology. TGA results from Figure 1c gave further evidence regarding the thermal stability analysis of ZIF-8 and ZIF-8-P under the nitrogen atmosphere. The change in the sample weights within the temperature range from 50 °C may have been due to the emission of H_2_O and the decomposition of ZIF-8 and ZIF-8-P. The sample of ZIF-8 revealed that the weight loss of the sample was 19.3 wt % at 600 °C, while the residue of ZIF-8-P remained at 44.6 wt % at 600 °C, which manifested the enhanced carbonization effect owing to the presence of PA. Thus, PA-modified ZIF-8 may have had an effective influence through the condensed phase in the fire test. In addition, according to the XPS survey spectra of the ZIF-8 and ZIF-8-P from Appendix A, the appearance of phosphorus element combined with the decreased dc (C/O) ratio further demonstrates the fact that PA had been introduced into ZIF-8.

FTIR spectroscopy is an important technique to research the functional groups of samples. Figure 1d shows the FTIR spectra of PMMA, PMMA/ZIF-8, and PMMA/ZIF-8-P. In the PMMA infrared spectrum, the absorption peak at 1727 cm^−1^ is the carbonyl absorption peak; the characteristic peaks corresponding to 2924 cm^−1^ and 2997 cm^−1^ are C-H bonds in alkyl groups [10]. In the infrared spectra of PMMA/ZIF-8, the absorption peaks at 3216 cm^−1^ and 2924 cm^−1^ are the stretching vibration absorption peaks of C-H bonds in methyl and imidazole rings, respectively; the absorption peak at 1576 cm^−1^ is the stretching vibration peak of the C=N of imidazole ring, and 1142 cm^−1^ is the contraction vibration peak of N-H in an imidazole ring [16,17]. In the infrared spectra of PMMA/ZIF-8-P, except for the absorption peaks of PMMA and ZIF-8, 753 cm^−1^, 2424 cm^−1^, and 1240 cm^−1^ corresponded to the absorption peaks of P-O-C, P-H, and P=O from PA [19,21]. Moreover, as exhibited in the XRD spectrum of ZIF-8-P from Appendix A, the existence of characteristic peaks indicated that the modification of PA did not cause the structural damage for ZIF-8. For the PMMA/ZIF-8-P composite, most diffraction peaks for ZIF-8-P disappeared or were overlapped by a broad diffraction peak of PMMA, suggesting that ZIF-8-P had been successfully incorporated into the PMMA matrix. SEM was used to investigate the inner structure of the composites to study the dispersion of fillers in the polymer matrix. Figure 1e,f show that, in the composite, ZIF-8 or ZIF-8-P had good interfacial adhesion with the PMMA matrix, which exhibited a uniformly dispersed morphology.

The mechanical properties of materials in microscopic aspects, such as the hardness and elastic modulus, are used to perform analysis via nanoindentation [22]. Figure 2a shows the loading and unloading curves of the PMMA and PMMA composite materials under the condition of a 2 mN load, and the hardness and elastic modulus values were computed by the Oliver–Pharr method. On the basis of the data in Table 1, the hardness of pure PMMA was 0.344 GPa, while the hardness of PMMA/ZIF-8 and PMMA/ZIF-8-P were 0.470 GPa and 0.602 GPa, respectively, both of which are higher than that of pure PMMA, agreeing well with the results of the reduced modulus (6.258 GPa for PMMA, 7.248 GPa for PMMA/ZIF-8 and 8.565 GPa for PMMA/ZIF-8-P). Because of the stronger interaction between ZIF-8-P and PMMA, the ability to resist the plastic deformation of PMMA/ZIF-8-P composites has been greatly strengthened. The thermomechanical properties of PMMA and PMMA composite materials were analyzed by a dynamic mechanical analyzer (DMA), giving the information of storage modulus (E′) within the measured temperature range. Figure 2b reveals that E′ varied with temperature for PMMA, PMMA/ZIF-8, and PMMA/ZIF-8-P. Below the glassy state region, the PMMA/ZIF-8 and PMMA/ZIF-8-P composites displayed a higher increase of E′ compared to that of PMMA. The enhancement effect of ZIF-8 or ZIF-8-P was due to the stronger non-covalent bond interactions between ZIF-8 or ZIF-8-P and PMMA, which limited the movement of the polymer matrix and resized the external forces to enhance the mechanical properties of the polymer.

The surface resistance values of the samples were measured for characterizing their antistatic properties [10]. The surface resistance of a plastic has a sufficient antistatic effort when the surface resistance is in the range of 10^8^–10^10^ Ω [23]. Figure 2c displays the surface resistance results for the PMMA, PMMA/ZIF-8, and PMMA/ZIF-8-P composites, which demonstrate that PMMA could obtain an adequate antistatic property by incorporating ZIF-8 and ZIF-8-P. The surface resistance of the ZIF-8-P decreased to 10^10^ Ω, which could reduce the electrostatic hazard during applications and avoid electrostatic charge accumulation. Figure 2d illustrates the antistatic mechanism of the PMMA/ZIF-8-P composite. The hydroxyl and phosphate groups from ZIF-8-P absorb moisture in the air, augmenting the hygroscopicity of the product, which could form a conductive film on the surface of PMMA. The molecular water layer promotes particle ionization in plastic products, thus endowing a reasonably sufficient antistatic property [23].

The thermal stability of the synthesized samples was examined by TGA. The TGA curves of the PMMA, PMMA/ZIF-8, and PMMA/ZIF-8-P composites under nitrogen and air atmospheres are shown in Figure 3a,b and the data are summarized in Table 1. The gas emissions from first mass loss can be attributed to the removal of volatile low molar mass components, and the following mass loss indicates the depolymerization initiated by the saturated chain ends and random main chain scission [4]. As displayed in the TGA curves and Table 1, the temperature at 50% weight loss for the PMMA/ZIF-8 and PMMA/ZIF-8-P composites enhanced in comparison with that of PMMA in nitrogen and air atmospheres. Besides, the PMMA/ZIF-8 composite generated 8.562 wt % and 4.140 wt % carbon residues under nitrogen and air conditions at 600 °C, respectively. Further, the carbon-rich residues of the PMMA/ZIF-8-P composite can reach 12.582 wt % and 5.648 wt % under nitrogen and air conditions at 600 °C, respectively, while almost no residue at the same temperature for pure PMMA remains. The improvement for the thermal stability reveals the superiority of the catalytic charring effect of ZIF-8 or ZIF-8-P.

To assess the flame retardance property of the PMMA and PMMA composites, the LOI and UL-94 vertical tests performed, which provided the information about the oxygen concentration to maintain material burning and the ability of the material to extinguish after being ignited, respectively [24]. With the addition of ZIF-8, the LOI for the composite grew to 21 ± 0.5% and a V-2 rating was obtained for the UL-94 vertical test. For the PMMA/ZIF-8-P composite, the LOI could reach 24 ± 0.5% and a V-1 rating was obtained for the UL-94 vertical burning test. These results are better than those of pure PMMA (LOI value: 17 ± 0.5%; no UL-94 rating), and even more efficient than those of most reported PMMA composites with flame retardants [25,26]. Furthermore, the flame retardancy was evaluated through the use of microscale combustion calorimetry (MCC), which offered rich information about the combustion properties of the materials [22]. The HRR curves are demonstrated in Figure 3c and the corresponding data are summarized in Table 1. It was found that the introduction of ZIF-8 or ZIF-8-P brought about a reduction of the peak heat release rate (PHRR) from 233.303 W/g for PMMA to 195.939 and 170.483 W/g for the corresponding composites, respectively. Moreover, in comparison to the 16.700 kJ/g of pure PMMA, the addition of ZIF-8 and ZIF-8-P decreased the total heat release (THR) values to 14.050 and 12.280 kJ/g, and the heat release capacity (HRC) values also reduced from 225.2 to 189.134 and 164.562 J/g/K, respectively. These results reveal the great benefits of ZIF-8 or ZIF-8-P for effectively reducing the heat hazards of PMMA.

Moreover, the pyrolysis products of PMMA and its composite materials were tested using TGA-FTIR, which gave the relationship between the intensity of the total released gas and time [27]. As demonstrated in Figure 3d, compared with pure PMMA, the total pyrolysis produced by PMMA/ZIF-8 and PMMA/ZIF-8-P composites all decreased significantly. The PMMA/ZIF-8-P composite produced the least pyrolysis products in the combustion process, due to the strong adsorption capacity and the inhibition of barrier resulting from the formation of char catalyzed by ZIF-8-P, which hindered the generation and transmission of pyrolysis products. Therefore, incorporating the ZIF-8 or ZIF-8-P generated a good benefit on decreasing non-heat hazards of PMMA.

In order to explore the flame retardance mechanism, SEM and LRS were used to examine the compositions and structures of residual carbon generated from the PMMA composites after combustion. Different from pure PMMA with no residual carbon, both PMMA/ZIF-8 and PMMA/ZIF-8-P could form char after combustion. As shown in the SEM images of Figure 4a,b, the remnant char of the PMMA/ZIF-8 sample possessed lots of fragile holes, while a compact and continuous char morphology could be obtained for the PMMA/ZIF-8-P sample. The reinforced char residue from the PMMA/ZIF-8-P sample may be present due to the existence of PA. As reported previously, phosphate groups can act as a highly efficient catalysts toward the formation of graphitized char and can self-decompose into a protective phosphorus-rich surface layer during a burning process [21]. The Raman spectra were further used to analyze the residual char structures (Figure 4c), which showed obvious G-band and D-band peaks around 1585 and 1350 cm^−1^, respectively. The former represented highly graphitic carbon structures, whereas the latter D-band peaks came from the disordered carbon or defective carbon [28]. As calculated, the intensity ratio (I_G_/I_D_) of PMMA/ZIF-8-P exceeded that of PMMA/ZIF-8 (0.4177 vs. 0.4001), demonstrating the higher graphitization level of char from the PMMA/ZIF-8-P composite, which is beneficial to blocking heat and gas transfer.

As for the analysis above, a possible fire safety mechanism for the ZIF-8-P in PMMA was put forward in Figure 4d. ZIF-8-P decomposes to generate metal-containing residue and amino and phosphoric acid compounds. On the one hand, during the combustion, polyphosphoric acid could form graphitized char and build a phosphorus-rich protective layer, thus impeding heat and mass transmission, oxygen permeation, and the release of combustible gases [22]. Moreover, a metal-containing residue (showed from TGA results) absorbs the pyrolysis products of a polymer and supports a catalytic oxidation reaction, slowing the emission of smoke and toxic gases [29], which is consistent with the above results regarding TGA-FTIR. On the other hand, incombustible gases (water, ammonia, etc.), produced from the degradation of organic ligands containing amino in ZIF-8-P, lead to a decrease in oxygen and inflammable gases [30]. One can make an assertion that the enhanced flame resistance and the reduced pyrolysis products can be ascribed to the superiority of the combinative effect of the catalytic charring, char reinforcement, and the dilution of incombustible gases from ZIF-8-P.

## 4. Conclusions

An efficient method for the enhancement of mechanical and fire safety for PMMA has been reported via introducing ZIF-8-P as additives. The PMMA/ZIF-8-P composite displayed remarkable mechanical and fire safety performance, with an enhancement of ca. 136.9% and 175.0% for the reduced modulus and hardness, respectively, as well as a reduction of ca. 73.1% for the PHRR, and an increase of ca. 41.2% for LOI and a lower amount of total pyrolysis products, which is superior to the results for pure PMMA and a PMMA/ZIF-8 composite. It can be deduced that based on the analysis of the carbon layer, combined with the above performance results, the improvement in performance here is due to stronger non-covalent bond interactions, hygroscopicity groups, adsorption capacity, and the barrier effect of the as-formed char during a burning process. Our results provide a new path to produce PMMA-based materials with superior mechanical and fire-safety properties.

## Figures and Tables

**Figure 1 polymers-14-04871-f001:**
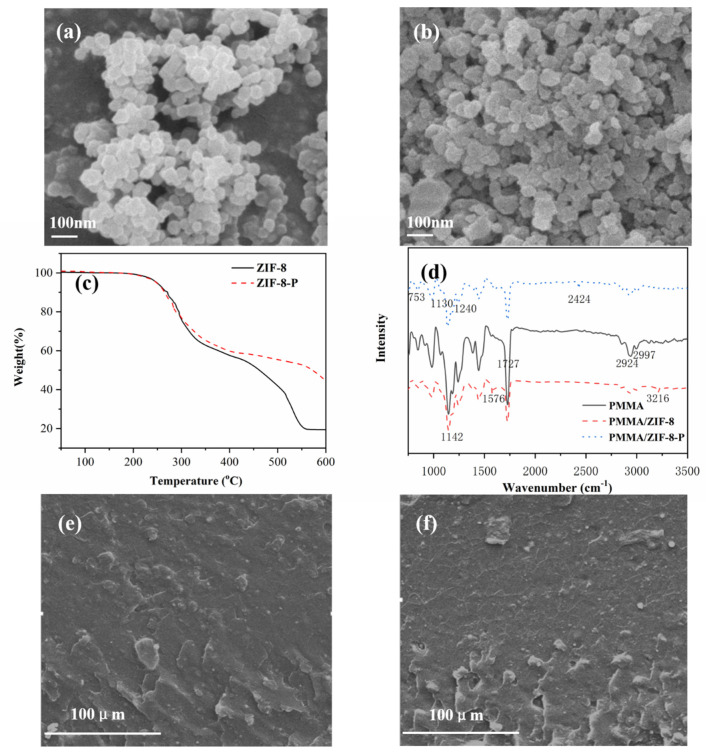
(**a**) SEM images of ZIF-8 and (**b**) ZIF-8-P. (**c**) TGA results of ZIF-8 and ZIF-8-P under a nitrogen atmosphere. (**d**) FTIR spectra of PMMA, PMMA/ZIF-8 and PMMA/ZIF-8-P. (**e**) SEM images of PMMA/ZIF-8 and (**f**) PMMA/ZIF-8-P composites.

**Figure 2 polymers-14-04871-f002:**
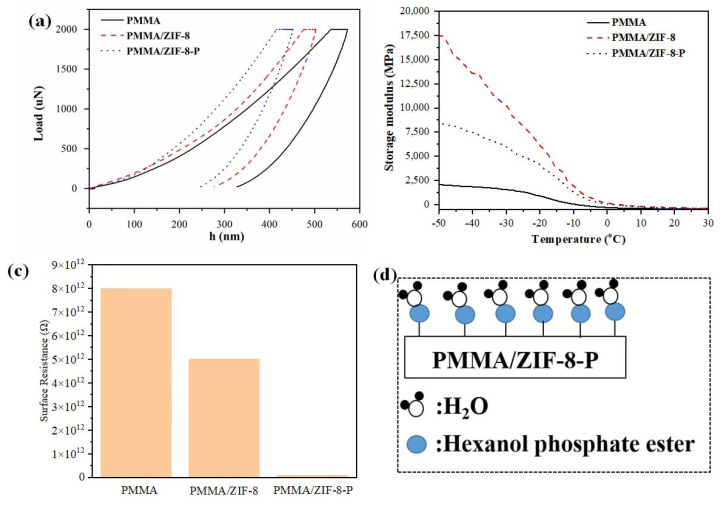
(**a**) The curves for load–displacement of the PMMA, PMMA/ZIF-8 and PMMA/ZIF-8-P composites. (**b**) Storage modulus gained from the DMA analysis of PMMA, PMMA/ZIF-8 and PMMA/ZIF-8-P composites. (**c**) Surface resistance of the PMMA, PMMA/ZIF-8 and PMMA/ZIF-8-P composites. (**d**) Antistatic mechanism of the PMMA/ZIF-8-P composite.

**Figure 3 polymers-14-04871-f003:**
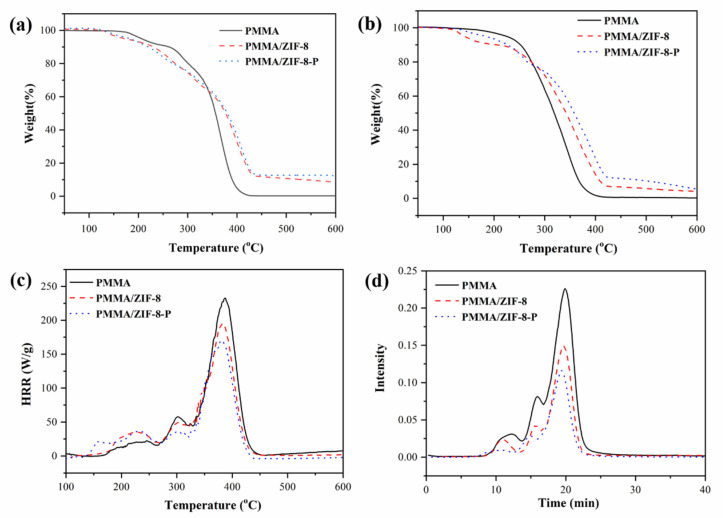
Thermogravimetric analysis (TGA) curves of PMMA, PMMA/ZIF-8, and PMMA/ZIF-8-P composites under (**a**) nitrogen and (**b**) air atmospheres at a heating rate of 20 °C/min. (**c**) Heat release rates (HRRs), along with the temperatures of the PMMA, PMMA/ZIF-8, and PMMA/ZIF-8-P composites in the microscale combustion calorimeter (MCC) test. (**d**) Intensity of total released gas during thermal degradation process of pure PMMA, PMMA/ZIF-8, and PMMA/ZIF-8-P composites in a nitrogen atmosphere with a heating rate of 10 °C/min.

**Figure 4 polymers-14-04871-f004:**
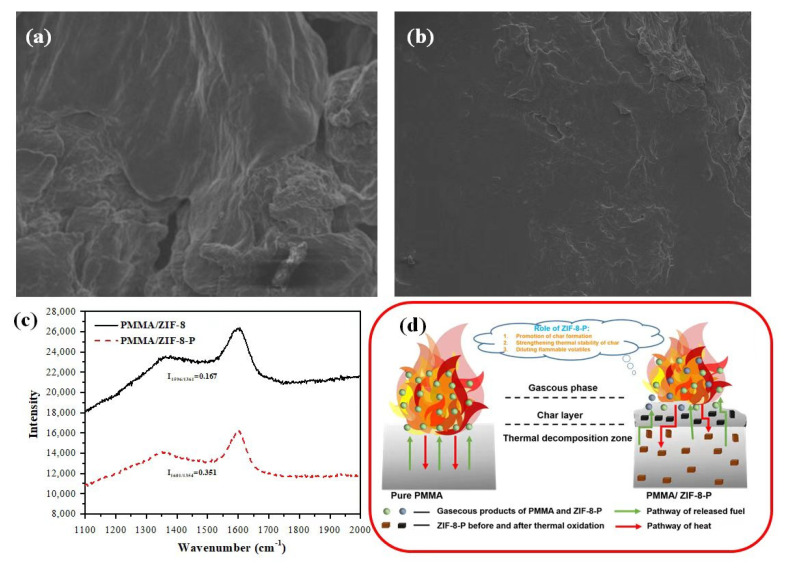
SEM of the char, including **the** (**a**) PMMA/ZIF-8 and (**b**) PMMA/ZIF-8-P composites. (**c**) Raman spectra for the carbon residue of the PMMA/ZIF-8 and PMMA/ZIF-8-P composites. (**d**) Schematic diagram of improving the flame retardance of the PMMA/ZIF-8-P composite.

**Table 1 polymers-14-04871-t001:** Nanoindentation results for the TGA and MCC data for the PMMA, PMMA/ZIF-8, and PMMA/ZIF-8-P composites.

Samples	Nanoindentation		TGA				MCC		
	Reduced Modulus (GPa)	Hardness (GPa)	Nitrogen Atmosphere		Air Atmosphere		PHRR (W/g)	HRC (J/g/K)	THR (kJ/g)
			T_-50_ (°C)	Residue at 700 °C (%)	T_-50_ (°C)	Residue at 700 °C (%)			
PMMA	6.258	0.344	356.474	0	318.607	0	233.303	225.200	16.700
PMMA/ZIF-8	7.248	0.470	377.148	8.562	343.231	4.140	195.939	189.134	14.050
PMMA/ZIF-8-P	8.565	0.602	380.524	12.582	358.437	5.648	170.483	164.562	12.280

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
