# Peer review of "Phosphorylated Metal–Organic Framework for Reducing Fire Hazards of Poly(Methyl Methacrylate)"

_polymers, 2022, doi:10.3390/polym14224871_

Round 1
Reviewer 1 Report
In this work, phosphorylated Zn-based metal-organic-framework (ZIF-8-P) was employed as an effective flame retardant and antistatic to reduce the fire risk of PMMA. PMMA is a versatile material that is practically used everywhere, thus the fire risks are an important aspect to address. Furthermore, the added antistatic properties are desirable for hazardous environments. The content of this article is in scope of this journal and is well written. I recommend the manuscript to undergo major revision to address the comments below:
1. In this article, only TGA and MCC have been conducted. Is there justification for not including standard cone calorimetry? MCC is not a fully reliable test as the samples are too small and the combustion is isolated from the material decomposition. Charring performance can also significantly differ
2. There is no examination of the gas emissions from TGA or cone.
3. How does the approach adopted in this article compare to other similar flame retardants for PMMA? It would be good to benchmark against other works in terms of flammability reduction.
4. The evidence to justify the proposed flame retardant mechanism are a bit weak. Especially the section on emissions of smoke and toxic gases. Most of the justifications are references and does not refer to results presented in this work. It is recommended to provide more analytical results from fire testing of the actual samples from this work.
5. The conclusion is too short and brief. Please try to include more details.
Reviewer 2 Report
The manuscript presents the influence of a Zn-based Metal-Organic-Framework (ZIF-8) and of corresponding phosphorylated (ZIF-8-P) in reducing the fire hazard of poly(methyl methacrylate) (PMMA). PMMA is widely used in various fields; however, it has the disadvantage of being highly flammable, which restricts the possibilities of its applications. The use of Metal-Organic-Frameworks (MOFs) and especially phosphorylated MOFs as flame retardant additives seems to be a promising method to reduce the flammability of PMMA.
Some remarks:
- The authors should explain why they chose the ZIF-8-P/PMMA ratio of 1:99 by weight for the preparation of PMMA/ZIF-8-P nanocomposite. Only one PMMA/ZIF-8-P sample was prepared and investigated and therefore a study of the influence of ZIF-8-P content on the properties of nanocomposites was not carried out.
- The authors should specify if there are physical interactions or chemical bonds between ZIF-8 and phytic acid in ZIF-8-P or if it is a physical mixture.
- Figures 1e, 1f, 4a and 4b, the magnification in the case of the SEM analysis should be given.
- Microscale combustion calorimeter (MCC) is a method that can determine several parameters: HRR (as a function of temperature and time), PHRR, THR, HRC, and char yield. However, the authors have only discussed HRR and PHRR parameters. HRC (heat release capacity) and THR (Total heat release) could also be considered.
- The reference concerning the ZIF-8 preparation should be given.
- The heating rate of the thermogravimetric analysis should be given.
The manuscript is well written, and the characterization of the investigated products was carried out using a large number of methods. It is interesting and deserves to be published in the journal Polymers.
Reviewer 3 Report
This work by Wang and colleagues describes the preparation of a phosphorylated PMMA/ZIF-8P composite and its potential application as flame retardant.
There are some points that the authors should address in any subsequent revison:
1. The English style should be revised in depth.
2. PXRD of the MOFs and composites must be performed. The authors should discuss the results obtained using this technique and incorporate the spectra in the manuscript (or in a supporting information containing all the spectra).
3. XPS should be used in order to characterize ZIF-8 and ZIF-8P.
4. The authors should consider to use AFM to further characterize their MOFs and composites.
5. Although the authors include a reference on LOI and UL-94 tests within the manuscript, they should incorporate some paragraphs explaining these tests. This information will be really useful to easily understand this important point of the manuscript without consulting an additional work.
Round 2
Reviewer 1 Report
The authors have sufficiently addressed my issues and it is recommended for publication.
Reviewer 3 Report
The current version of the manuscript is suitable for publication in Polymers.